# Position: Explanation Stability Is a Property of the Model–Method Pair, Not the Model

Kabilan Elangovan [1]   Daniel Ting [1]

## Abstract

This position paper argues that explanation stability claims are scientifically invalid without cross-method validation. Just as statistical significance requires specifying the test statistic, stability must be validated across multiple attribution paradigms or explicitly scoped to a single method's computational objective. In controlled chest X-ray experiments, DenseNet201, ResNet50V2, and InceptionV3 achieve >99% AUC but exhibit reversed stability rankings across attribution methods. LayerCAM ranks InceptionV3 highest (IoU 0.777), while Grad-CAM++ favors DenseNet201, reducing InceptionV3's score by 17.3%. These findings establish that explanation stability is an emergent property of the model–method pair, not an intrinsic model trait. We argue that explanation-based claims should be validated across multiple attribution methods and urge that regulatory submissions explicitly specify attribution operators to avoid illusory safety assurances.

## 1. Introduction: Why Stability Must Be Scoped to the Attribution Method

**Position Statement:** Claims of explanation stability cannot be scientifically justified without explicitly scoping the evaluation to the attribution method under consideration—just as claims of statistical significance require specification of the test statistic. We do not argue that explanations must agree across methods; rather, analyses based on a single attribution method cannot support claims of *general* stability beyond that method's specific computational objective and inductive biases (Krishna et al., 2024; Han et al., 2022).

[1]Singapore Health Services and Singapore Eye Research Institute, Singapore. Correspondence to: Kabilan Elangovan <kabilan.elangovan@singhealth.com.sg>.

*Proceedings of the 43rd International Conference on Machine Learning*, Seoul, South Korea. PMLR 306, 2026. Copyright 2026 by the author(s).

Transfer learning dominates contemporary medical image classification pipelines (Raghu et al., 2019; Kolesnikov et al., 2020), yet the stability of visual explanations across transfer learning and fine-tuning is rarely examined. Consider a convolutional model that correctly classifies COVID-19 pneumonia and highlights bilateral ground-glass opacities *according to* Grad-CAM (Selvaraju et al., 2017). After task-specific fine-tuning, classification performance remains unchanged, but the resulting attribution maps emphasize different image regions. Both predictions are correct, yet the apparent evidential basis of the decision—*as measured by the attribution method*—has shifted. Such sensitivity of attribution maps to parameterization and optimization has been widely observed even in the absence of accuracy degradation (Adebayo et al., 2018; Ghorbani et al., 2019; Kindermans et al., 2019).

This raises a fundamental question: does this instability reflect a genuine change in the model's internal representations, or variability induced by the attribution method itself? Without cross-method validation, attribution-based stability analyses risk conflating method-specific behavior with model behavior.

### 1.1. Why Current Practice Fails

Current XAI evaluation practice is limited by a recurring pattern: explanation methods are evaluated in isolation, stability or faithfulness is reported under a single technique, and these findings are implicitly generalized to explanation quality more broadly. Such generalization is methodologically unjustified, as empirical evidence consistently demonstrates substantial disagreement and context dependence across attribution methods.

**Explanation disagreement is pervasive.** A large practitioner study reports that 84% of users encounter conflicting explanations for identical model predictions, with no standardized framework to resolve such disagreements (Krishna et al., 2024). Common methods such as LIME, KernelSHAP, and Integrated Gradients often yield contradictory attributions, leaving practitioners to rely on ad hoc heuristics.

**Relative method rankings are unstable.** Using a relative

attribution ranking framework, Duan et al. show that the performance ordering of eight attribution methods varies markedly across architectures, datasets, and evaluation settings (Duan et al., 2024). Rankings derived under restricted conditions fail to generalize to heterogeneous scenarios, undermining claims of method-level consistency.

**Large-scale empirical validation confirms pervasive instability.** The LATEC benchmark (Klein et al., 2024)—evaluating 17 methods across 20 metrics in 7,560 combinations—demonstrates that conflicting metrics produce unreliable rankings, reinforcing that no single method-metric pair provides universal ground truth.

**No universal ground truth exists for explanation correctness.** The Quantus toolkit demonstrates that evaluation outcomes are highly sensitive to metric choice and parameterization across more than 35 commonly used measures, precluding a single definitive notion of explanation quality (Hedström et al., 2023b).

**Clinical benchmarks expose a persistent gap.** In chest X-ray interpretation, all evaluated saliency methods—including Grad-CAM—perform substantially worse than radiologists at localizing clinically relevant findings, with gaps varying across pathologies (Saporta et al., 2022).

These limitations reflect fundamental differences in the mathematical objectives of attribution methods rather than mere measurement noise. Gradient-based approaches encode distinct notions of importance, from local pixel sensitivity to globally aggregated relevance, which are not mathematically equivalent (Ancona et al., 2018). Consequently, agreement across methods should not be assumed, and stability claims must be explicitly scoped to the attribution technique employed.

### 1.2. Why This Matters for Medical AI

Medical AI deployment faces increasing regulatory scrutiny (FDA and Health Canada and MHRA, 2021; FDA, 2025), with substantial variability in explainability across FDA-cleared systems (McNamara et al., 2024). Method-dependent stability complicates regulatory evaluation of explanation reliability. Clinical studies further underscore the implications of this ambiguity: clinicians often find technically sophisticated explanations, such as Shapley-based attributions, misaligned with clinical reasoning and decision-making needs (Bienefeld et al., 2023), while prior work cautions that inconsistent or poorly contextualized explanations may mislead clinical judgment rather than support it (Babic et al., 2021; Ghassemi et al., 2021).

### 1.3. Our Position and Contributions

We argue that **the field cannot justify general explanation stability claims from single-method evaluation**. Our

position rests on three pillars:

1. **Empirical demonstration of method-dependency:** Through controlled experiments quantifying semantic drift—visual evidence transformation during transfer learning and fine-tuning—we show that LayerCAM and Grad-CAM++ produce systematically contradictory stability rankings across DenseNet201, ResNet50V2, and InceptionV3 despite equivalent predictive performance.
2. **True-positive filtering for unconfounded analysis:** By restricting analysis to samples correctly classified in both training phases, we isolate explanation evolution independent of prediction quality changes, eliminating the confound that instability reflects error correction.
3. **Architectural mechanism analysis:** We show that dense connectivity promotes cross-method stability (DenseNet), multi-scale pathways induce layer-dependent stability (InceptionV3), and residual shortcuts enable method-sensitive reorganization (ResNet).

## 2. Background: The Attribution Method Problem

### 2.1. Gradient-Based Attribution Fundamentals

**Grad-CAM** (Selvaraju et al., 2017) uses global average pooling of gradients, weighting entire feature maps uniformly. **Grad-CAM++** (Chattopadhay et al., 2018) incorporates higher-order gradients for multi-instance localization through adaptive weighting. **LayerCAM** (Jiang et al., 2021) applies pixel-wise gradients preserving fine-grained spatial detail. These are not interchangeable—they solve fundamentally different mathematical objectives (Ancona et al., 2018).

### 2.2. The Sanity Check Crisis

Adebayo et al.'s seminal work (Adebayo et al., 2018) revealed that some attribution methods (Guided Backpropagation) function as edge detectors independent of model parameters or training data. Methods failing architecture-dependent sanity checks produce explanations unrelated to learned representations. Critically, **passing sanity checks under one method doesn't guarantee validity under another**.

### 2.3. The Disagreement Problem

Krishna et al. (Krishna et al., 2024) quantified practitioner-encountered disagreements: LIME, KernelSHAP, and Integrated Gradients frequently contradict across four real-world datasets. The ROAR benchmark (Hooker et al., 2019) showed many popular methods produce importance estimates no better than random, with performance rankings varying by dataset.

Recent work establishes theoretical limits: no single attribution method can universally approximate model behavior faithfully (Han et al., 2022). A no-free-lunch theorem for explanations implies that method choice inevitably introduces bias.

### 2.4. Medical Imaging Evaluation Gaps

Saporta et al. (Saporta et al., 2022) created the first human benchmark for chest X-ray saliency, revealing all seven tested methods (including Grad-CAM) performed significantly worse than radiologists. Arun et al. (Arun et al., 2021) demonstrated that InceptionV3 saliency maps showed higher utility than DenseNet-121, with XRAI achieving highest repeatability on InceptionV3 but different performance on DenseNet-121. **Reproducibility across architectures was consistently low.**

### 2.5. Aggregation-Based Solutions

Aggregation frameworks (Pirie et al., 2023; Schwarzschild et al., 2023; Kazmierczak et al., 2025) address disagreement by weighting attributions across explainers or training for consensus. These approaches treat method-dependency as a problem to resolve rather than characterize. Our position differs: we argue practitioners need to identify which architectures exhibit method-robust stability *before* deployment, not aggregate disagreeing methods post-hoc.

### 2.6. Gap Addressed

No prior work systematically quantifies explanation stability across architectures *and* attribution methods during fine-tuning with true-positive filtering. We reveal method-dependent stability rankings that challenge single-method evaluation, establishing cross-method validation as necessary for trustworthy clinical deployment.

## 3. Evidence: Semantic Drift Reveals Method-Dependent Stability

### 3.1. Experimental Design

We conduct controlled five-class chest X-ray classification (Normal, Pneumonia, Tuberculosis, COVID-19, Lung Opacity) on 3,354 test samples with severe class imbalance (4.2–35.8%). Three ImageNet-pretrained architectures—DenseNet201 (18.3M parameters), ResNet50V2 (23.6M), and InceptionV3 (21.8M)—are trained in two phases: transfer learning (epochs 1–10, frozen backbone) and fine-tuning (epochs 11–20, unfrozen layers). We compare epoch 8 (transfer-learning plateau, AUC 0.971–0.974) with epoch 19 (fine-tuning convergence, AUC ≥0.991) to maximize semantic drift contrast.

**Attribution Methods:** LayerCAM (pixel-wise gradients) and Grad-CAM++ (higher-order adaptive weighting) are applied to penultimate convolutional layers. Maps are normalized to [0,1] and thresholded at $\tau = 0.2$. Explanation changes are quantified using four complementary metrics: overlap IoU (primary), spatial displacement, pattern correlation, and concentration change.

**True-Positive Filtering:** To isolate explanation evolution from classification effects, sample $(x_i, y_i)$ is included only if correctly classified by *all three architectures* in *both training phases*. This eliminates confounding from accuracy changes, ensuring measured drift reflects genuine explanation evolution rather than prediction corrections. Of 3,354 test samples, 2,430 (72.5%) satisfy this criterion. Results are weighted by inverse class frequency (tuberculosis: 0.528; lung opacity: 0.062) to correct for imbalance. Full details in Appendix A.

### 3.2. Semantic Drift Metrics

Four complementary reference-free metrics quantify explanation transformation:

**Spatial Displacement** measures center-of-mass movement:

$$\Delta_{\text{spatial}} = \frac{||\text{CoM}(\tilde{A}_{\text{TL}}) - \text{CoM}(\tilde{A}_{\text{FT}})||_2}{\sqrt{h^2 + w^2}} \quad (1)$$

**Overlap IoU** (primary metric) quantifies evidential consistency:

$$\text{IoU} = \frac{|M_{\text{TL}} \cap M_{\text{FT}}|}{|M_{\text{TL}} \cup M_{\text{FT}}|} \quad (2)$$

where $M = \mathbb{1}[\tilde{A} > \tau]$ are binary masks.

**Pattern Correlation** captures continuous similarity via Pearson coefficient. **Concentration Change** quantifies attention sharpening via Shannon entropy difference.

Inverse frequency weighting ensures tuberculosis (4.2%, weight 0.528) contributes proportionally to lung opacity (35.8%, weight 0.062):

$$\bar{\Delta}_{\text{weighted}} = \sum_{c=1}^{K} \tilde{w}_c \cdot \frac{1}{N_c} \sum_{i \in \mathcal{D}_c} \Delta(x_i) \quad (3)$$

### 3.3. Results: Method-Dependent Rankings

Table 2 and Figure 1 reveal the core finding: **attribution method choice determines apparent architecture stability**.

**LayerCAM findings:** InceptionV3 (IoU=0.777 ± 0.128) demonstrates superior stability, outperforming DenseNet201 (0.699 ± 0.171) by 11.2% and ResNet50V2 (0.519 ± 0.154) by 49.7%.

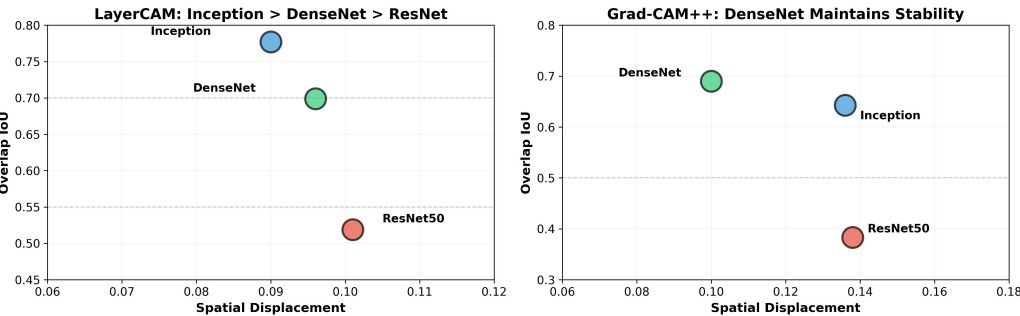

*Figure 1.* **Method-dependent stability rankings reveal complete architectural reversal. Left (LayerCAM):** InceptionV3 (IoU=0.777) outperforms DenseNet201 (0.699) and ResNet50V2 (0.519). **Right (Grad-CAM++):** DenseNet201 (0.690) now leads InceptionV3 (0.643) and ResNet50V2 (0.383). InceptionV3 exhibits 17.3% method-dependent degradation; DenseNet maintains 1.3% cross-method consistency. Spatial displacement increases substantially under Grad-CAM++ for InceptionV3 (+51%) and ResNet (+37%), while DenseNet remains stable (+4%), demonstrating orthogonality between spatial and structural consistency.

*Table 1.* Test performance at Epoch 19 (fine-tuned). All architectures achieve >99% AUC with comparable accuracy and F1 scores, demonstrating equivalent predictive capability despite divergent explanation stability.

| Architecture | AUC | Accuracy | F1-Score |
|---|---|---|---|
| DenseNet201 | 0.995 | 0.936 | 0.935 |
| ResNet50V2 | 0.998 | 0.973 | 0.959 |
| InceptionV3 | 0.998 | 0.973 | 0.964 |

**Grad-CAM++ reveals ranking reversal:** DenseNet201 (0.690 ± 0.169) now leads InceptionV3 (0.643 ± 0.172), while ResNet50V2 (0.383 ± 0.174) exhibits catastrophic 26.2% degradation from LayerCAM.

**DenseNet as method-agnostic:** Only DenseNet demonstrates cross-method robustness (0.699→0.690, 1.3% variation). This stability indicates dense connectivity promotes coherent explanation evolution regardless of gradient paradigm.

**InceptionV3's hidden fragility:** 17.3% performance degradation reveals multi-scale processing produces stable pixel-wise patterns (LayerCAM) but volatile scale-pathway interactions (Grad-CAM++). High single-method performance masks method-dependent vulnerability.

**ResNet's universal failure:** Poor performance across both methods (0.519→0.383) plus extreme concentration (−0.516) suggests residual shortcuts enable dramatic reorganization without structural coherence.

**Spatial-structural orthogonality:** Spatial displacement remains low under LayerCAM (0.090–0.101) but increases dramatically for ResNet (0.138, +37%) and InceptionV3 (0.136, +51%) under Grad-CAM++, while DenseNet maintains stability (0.100, +4%). Models preserve anatomical localization ("where") while reorganizing evidential

structure ("how").

Statistical validation (Appendix Table 4) confirms all architecture comparisons achieve $p < 0.001$ significance, with InceptionV3's cross-method difference showing Cohen's $d = 1.47$ (large effect).

### 3.4. Per-Class Analysis Reinforces Method-Dependency

Appendix Tables 7 and 8 summarize pathology-specific explanation stability across attribution methods. For InceptionV3 on normal cases, overlap IoU decreases from 0.806 under LayerCAM to 0.622 under Grad-CAM++, corresponding to a 22.8% reduction. A more pronounced decline is observed for COVID-19, where IoU drops from 0.720 to 0.404 (43.9% reduction). Similar method-dependent shifts are observed across other pathologies, including pneumonia (0.774 to 0.606; 21.7%), opacity (0.767 to 0.725; 5.5%), and tuberculosis (0.775 to 0.688; 11.2%).

These results demonstrate that **per-pathology explanation stability rankings are highly sensitive to attribution method choice**, even when model architecture and training protocol are held constant.

## 4. Why This Invalidates Current Practice

### 4.1. Single-Method Evaluation Cannot Support General Stability Claims

Our findings demonstrate that **explanation stability is an interaction between architecture and attribution method, not an intrinsic model property**. Researchers evaluating stability under LayerCAM would conclude that InceptionV3 equals or exceeds DenseNet201 (IoU 0.777 vs 0.699). Those using Grad-CAM++ would conclude that DenseNet201 substantially outperforms InceptionV3

*Table 2.* Method-dependent semantic drift metrics (weighted, $N = 2430$ true-positive test images). InceptionV3 dominates LayerCAM (0.777 IoU, bold) but DenseNet achieves cross-method robustness (1.3% change vs InceptionV3's 17.3% degradation). ResNet exhibits universal instability (26.2% collapse). Standard deviations indicate substantial inter-sample variability; concentration patterns diverge between methods.

| Method | Architecture | Spatial Disp | Overlap IoU | Pattern Corr | Conc Change |
|---|---|---|---|---|---|
| LayerCAM | DenseNet201 | $0.096 \pm 0.074$ | $0.699 \pm 0.171$ | $0.368 \pm 0.337$ | $-0.050 \pm 0.136$ |
| | ResNet50V2 | $0.101 \pm 0.062$ | $0.519 \pm 0.154$ | $0.403 \pm 0.285$ | $-0.136 \pm 0.130$ |
| | InceptionV3 | $0.090 \pm 0.058$ | $\mathbf{0.777} \pm 0.128$ | $0.220 \pm 0.465$ | $-0.024 \pm 0.077$ |
| Grad-CAM++ | DenseNet201 | $0.100 \pm 0.073$ | $\mathbf{0.690} \pm 0.169$ | $0.345 \pm 0.350$ | $-0.049 \pm 0.172$ |
| | ResNet50V2 | $0.138 \pm 0.085$ | $0.383 \pm 0.174$ | $0.506 \pm 0.246$ | $-0.516 \pm 0.516$ |
| | InceptionV3 | $0.136 \pm 0.073$ | $0.643 \pm 0.172$ | $0.386 \pm 0.423$ | $+0.275 \pm 0.303$ |

(0.690 vs 0.643). Both conclusions cannot simultaneously be correct about the models' *general* stability—yet both are valid within their respective computational frameworks.

This discrepancy is not a measurement precision issue—it reflects **fundamentally different computational objectives**. Pixel-wise gradient aggregation (LayerCAM) emphasizes localized feature importance, while globally pooled higher-order gradients (Grad-CAM++) capture scale-pathway interactions (Ancona et al., 2018). InceptionV3's parallel convolutional pathways can update semi-independently during fine-tuning, yielding stable local activations but volatile cross-scale integration.

The Meta-Rank findings (Duan et al., 2024) generalize this phenomenon: across multiple attribution methods and architectures, evaluation rankings systematically diverge under heterogeneous criteria. Our contribution extends this insight temporally—**stability rankings reverse even after predictive performance converges**—demonstrating that single-method assessments cannot support claims of general explanation stability.

### 4.2. Implications for Medical AI Deployment

**Regulatory ambiguity:** Recent FDA guidance requires explainability proportional to clinical risk (FDA, 2025), yet provides no framework for resolving attribution-method disagreement. If two FDA-cleared devices differ only in explanation method, they may produce contradictory clinical rationales despite identical predictions. How should clinicians interpret such divergence?

**Architecture selection is underspecified:** Current practice selects architectures primarily via cross-validation accuracy. Our results show that equivalent predictive performance ($>99\%$ AUC, Table 1) masks fundamental differences in explanation behavior. DenseNet201 exhibits strong cross-method robustness ($0.699 \rightarrow 0.690$ IoU; 1.3% change), whereas InceptionV3 shows pronounced method dependency ($0.777 \rightarrow 0.643$; 17.3% degradation)—a clinically meaningful distinction invisible to accuracy-only evaluation.

**Benchmark gaming:** Attribution-method choice can be selectively exploited to demonstrate apparent stability. Reporting LayerCAM results for InceptionV3 (IoU = 0.777) while omitting Grad-CAM++ results (IoU = 0.643) creates unjustified confidence in explanation robustness. This risk is not hypothetical—one in three XAI studies rely exclusively on anecdotal evaluation (Nauta et al., 2023), providing ample opportunity for selective reporting.

### 4.3. Architectural Mechanisms

**DenseNet201 exhibits greater cross-method consistency** potentially because dense connectivity—where each layer receives inputs from all preceding layers—encourages broader feature reuse and progressive refinement, producing representations that appear more stable across both first-order and higher-order gradient formulations (Huang et al., 2017). **InceptionV3 demonstrates stronger method-dependency**, plausibly due to its parallel multi-scale pathways (Szegedy et al., 2016), where different attribution operators may emphasize distinct within-pathway versus cross-pathway interactions during fine-tuning. **ResNet50V2 shows comparatively unstable behavior** across methods, potentially reflecting the flexibility introduced by residual identity mappings, which permit localized feature redistribution during optimization (He et al., 2016; Li et al., 2018). While these interpretations remain hypothesis-generating rather than definitive mechanistic explanations, they suggest that architectural inductive biases may influence the degree of cross-method explanation stability, even when predictive performance is comparable.

## 5. Alternative Views

We address three credible alternative positions challenging the need for cross-method validation in XAI evaluation.

## 5.1. Alternative View 1: Stability Under a Single Well-Chosen Method Suffices

**Position:** If practitioners select the attribution method most appropriate for a given architecture and clinical task, single-method evaluation provides sufficient stability guarantees.

**Our response:** This position assumes (a) principled criteria exist for matching methods to architectures, and (b) method-specific stability implies general trustworthiness. Our evidence contradicts both. No consensus framework exists for method–architecture matching; practitioners rely on ad hoc heuristics (Krishna et al., 2024), and Grad-CAM's dominance in medical imaging reflects historical convention rather than validated architectural suitability. More critically, method-specific stability does not imply robustness. InceptionV3 exhibits high LayerCAM stability (IoU = 0.777) yet degrades substantially under Grad-CAM++ (IoU = 0.643), indicating explanation behavior contingent on computational implementation rather than intrinsic model properties. Recent FDA work on explainability evaluation (Lago et al., 2025) identifies consistency—stability to input perturbations—as a prerequisite for trust, but evaluates consistency within a single attribution method. Our results extend this logic: a model stable to input noise but unstable to attribution method choice exhibits the same fragility these frameworks aim to detect. Emerging regulatory guidance (FDA, 2025) emphasizes robustness and transparency across settings—method-contingent stability fails this requirement.

**Thus, single-method stability cannot support general trustworthiness claims without explicit scoping to that method's computational objective.**

## 5.2. Alternative View 2: Explanation Instability Is Acceptable If Predictive Performance Remains Stable

**Position:** For deployment, predictive accuracy is sufficient. If fine-tuning maintains >99% AUC (Table 1), explanation evolution is clinically irrelevant.

**Our response:** This view treats explainability as a debugging aid rather than a deployment requirement. In contrast, FDA guidance (FDA, 2025), medical AI position papers (Ghassemi et al., 2021; Babic et al., 2021), and clinician surveys (Bienefeld et al., 2023) identify trustworthy explanations as a prerequisite for clinical adoption. Unstable explanations undermine clinician trust even when predictions remain correct. Highlighting different anatomical regions for the same pathology—across training phases or attribution methods—signals inconsistency in model reasoning. The Saporta et al. benchmark (Saporta et al., 2022) al-

ready demonstrates that attribution methods underperform human explanations; additional method dependence further erodes confidence. Importantly, our true-positive-filtered analysis shows instability arises *despite maintained correctness*. This reflects reorganization of internal representations rather than error correction (Raghu et al., 2019). When multiple evidential pathways yield equivalent accuracy, interpretability-critical deployment should favor architectures with stable explanatory pathways.

**Predictive performance alone therefore cannot justify claims of explanation stability; both dimensions require independent validation.**

## 5.3. Alternative View 3: Attribution Methods Measure Different Constructs—Disagreement Is Expected and Informative

**Position:** Different attribution methods formalize distinct notions of explanation. Disagreement is expected and provides complementary insights into model behavior (Ancona et al., 2018).

**Our response:** We agree in part—but this strengthens our position. If methods capture fundamentally different constructs, then single-method evaluation cannot claim general explanation stability, and multi-method validation becomes essential to characterize behavior comprehensively. Recent frameworks attempt to address this through aggregation: AGREE (Pirie et al., 2023) weights attributions by explainer confidence, PASTA (Kazmierczak et al., 2025) aligns explanations with human preferences, and PEAR (Schwarzschild et al., 2023) trains for explainer consensus. However, these solutions presume that consensus is always desirable and achievable.

For clinical deployment, however, complementary views must yield actionable guidance. If LayerCAM suggests InceptionV3 is stable while Grad-CAM++ suggests instability, aggregation produces a composite score—but which conclusion should govern deployment? **Our contribution is orthogonal**: rather than aggregating disagreeing methods post-hoc, we identify which architectures exhibit method-robust stability *before* deployment. DenseNet201's 1.3% cross-method variation versus InceptionV3's 17.3% degradation represents an architectural design principle for explanation-critical systems. For clinical deployment, however, complementary views must yield actionable guidance. If LayerCAM suggests InceptionV3 is stable while Grad-CAM++ suggests instability, which conclusion should govern deployment? Absent a meta-framework for resolving such conflicts which current practice lacks method diversity produces ambiguity rather than insight.

**Our position follows directly: multi-method evaluation**

is necessary *because* methods measure different constructs. Claiming "explanation stability" without specifying attribution method is analogous to claiming "statistical significance" without specifying the test.

# 6. Call to Action

We propose four concrete actions to address the method-dependency crisis:

## 6.1. Action 1: Mandatory Cross-Method Validation in XAI Research

**Who:** ICML, NeurIPS, ICLR, medical imaging venues (MICCAI, Medical Image Analysis, Radiology: AI)

**What:** Require XAI papers evaluating explanation stability, faithfulness, or trustworthiness to report results under *at least two attribution methods with different mathematical foundations* (e.g., pixel-wise vs. globally-pooled gradients). Papers claiming architecture-specific advantages must demonstrate cross-method robustness.

**How:** Update review guidelines to explicitly assess multi-method validation. Meta-review templates should include: "Does the paper evaluate explanations under multiple attribution paradigms? If not, do the claims appropriately caveat method-dependency?"

Existing frameworks already support systematic evaluation: OpenXAI (Agarwal et al., 2022) provides quantitative benchmarks for comparing explanation methods; Quantus (Hedström et al., 2023b) consolidates 30+ evaluation metrics across multiple assessment dimensions; and MetaQuantus (Hedström et al., 2023a) explicitly addresses disagreement among evaluation metrics. The infrastructure exists—policy must mandate its use.

## 6.2. Action 2: Regulatory Frameworks Must Fix Attribution Methods Before Certification

**Who:** FDA, European Medicines Agency (EMA), health technology assessment bodies

**What:** Medical AI devices claiming explainability should specify the attribution method in regulatory submissions. Post-market surveillance should monitor whether deployed methods match certified methods. Method changes require re-validation.

**Why:** Current FDA clearances show notable variability in explainability (McNamara et al., 2024) without standardization. If a device achieves clearance using Grad-CAM but hospitals deploy LayerCAM (or manufacturers switch methods post-clearance), clinicians receive explanations not validated by regulators.

**Precedent:** FDA's algorithm locking requirements for SaMD extend naturally to attribution methods. The 2025 AI/ML guidance (FDA, 2025) on lifecycle management supports attribution method specification in change control plans.

**FDA evaluation framework development:** Recent work by FDA-affiliated researchers (Lago et al., 2025) proposes a structured framework for evaluating explainability features across four dimensions: *consistency* (stability of explanations under input perturbations), *plausibility* (alignment with ground truth where available), *fidelity* (alignment with underlying model mechanisms), and *usefulness* (impact on clinician performance). This framework directly addresses a recognized evaluation gap in explainable AI by moving beyond single-metric assessments toward multi-dimensional characterization of explanation behavior in clinically relevant settings. Notably, the consistency criterion operationalizes robustness by assessing whether explanations remain stable under realistic input variations, a concern closely related to attribution method dependence. From a regulatory science perspective, extending such multi-dimensional evaluation to explicitly account for attribution method variability would be a logical progression, as changes in attribution methodology introduce behavioral variation that is not observable under single-method evaluation alone.

## 6.3. Action 3: Architecture Selection Should Prioritize Method-Robust Stability

**Who:** Medical AI developers, hospital IT procurement teams

**What:** When multiple architectures achieve equivalent predictive performance, prioritize those demonstrating superior cross-method explanation stability. Architecture selection should explicitly incorporate robustness to attribution-method choice, not accuracy alone.

**Evidence-based recommendation:** Our results identify DenseNet201 as method-robust (LayerCAM IoU = 0.699; Grad-CAM++ IoU = 0.690; 1.3% difference), whereas InceptionV3 exhibits pronounced method dependence (0.777 vs. 0.643; 17.3% degradation). Despite equivalent predictive performance (AUC > 0.99), these architectures differ substantially in explanation reliability. For explanation-critical deployment, DenseNet201's consistent behavior across attribution frameworks represents a clinically meaningful advantage invisible to accuracy-based selection.

**Design principle:** For explanation-critical medical AI, architectures enforcing global feature integration may provide more method-robust explanations than those relying on parallel processing or residual bypassing, even when predictive performance is equivalent.

### 6.4. Action 4: XAI Benchmark Standards Should Penalize Method-Sensitive Rankings

**Who:** Benchmark developers (OpenXAI (Agarwal et al., 2022), medical imaging benchmarks)

**What:** Establish "method-robustness" as a first-class evaluation dimension. Benchmark leaderboards should report not only per-method performance but also cross-method consistency. Penalize models achieving high performance under one method but poor performance under others.

**Implementation:** Compute cross-method variance for each model:

$$\sigma^2_{\text{method}}(M) = \frac{1}{|\mathcal{A}|} \sum_{a \in \mathcal{A}} (S_{M,a} - \bar{S}_M)^2 \qquad (4)$$

where $S_{M,a}$ is model $M$'s stability score under attribution method $a$, and $\mathcal{A}$ is the set of methods. Models with high $\sigma^2_{\text{method}}$ demonstrate fragile, method-contingent behavior.

**Existing infrastructure supports implementation.** LATEC (Klein et al., 2024) and PASTA (Kazmierczak et al., 2025) provide infrastructure requiring minimal extension to report cross-method variance alongside per-method performance. These frameworks require minimal extension to report cross-method variance as a first-class evaluation dimension.

**Precedent:** Adversarial robustness benchmarks (Robust-Bench) already penalize models optimized for clean accuracy but fragile to perturbations. Method-robustness extends this principle: explanations should be robust to computational implementation details.

## 7. Scope and Extensions

Our analysis evaluates two gradient-based attribution methods. Extending this framework to perturbation-based (Lundberg & Lee, 2017), path-based (Sundararajan et al., 2017), and attention-based approaches would further assess whether method-dependency persists across fundamentally different explanation paradigms, though theoretical no-free-lunch results (Han et al., 2022) suggest such variability is unlikely to be fully eliminated. We evaluate three CNN architectures; future work should investigate whether similar ranking reversals emerge in Vision Transformers (Raghu et al., 2021) and foundation models, where attention mechanisms may alter cross-method stability dynamics. Our work characterizes method-dependency in the context of architecture selection, but future studies should examine whether method-robust architectures also support more reliable explanation aggregation (Pirie et al., 2023), or whether architectural and aggregation-based approaches address complementary aspects of explanation variability. Importantly, our metrics quantify explanation consistency

rather than clinical correctness. Integrating expert radiologist annotations (Saporta et al., 2022) would strengthen assessment of human-aligned validity. While our experiments focus on medical imaging, the broader argument applies to safety-critical machine learning domains where explanation trustworthiness influences deployment decisions.

## 8. Conclusion

**Claims of general explanation stability cannot be scientifically justified without cross-method validation.** Through controlled experiments quantifying semantic drift from transfer learning to fine-tuning, we demonstrate that apparent architecture stability depends critically on attribution method choice. LayerCAM suggests DenseNet $\approx$ InceptionV3 > ResNet, whereas Grad-CAM++ reveals DenseNet > InceptionV3 > ResNet, with InceptionV3 exhibiting 17.3% cross-method degradation. This method-dependency persists despite equivalent predictive performance (>99% AUC), establishing explanation stability and classification accuracy as orthogonal dimensions. DenseNet201 demonstrates the greatest cross-method consistency (1.3% variation), suggesting that dense connectivity may promote more coherent explanation evolution across attribution operators.

**Our position challenges current evaluation practice:** researchers cannot claim "explanation stability" as a general model property without either (a) validating findings across attribution methods with distinct computational foundations, or (b) explicitly constraining conclusions to the specific explanation operator used. Similar to statistical inference requiring specification of the underlying test, explanation-based claims require explicit conditioning on the attribution paradigm used for evaluation.

We advocate for cross-method validation as a necessary consideration in XAI evaluation, particularly in safety-critical settings where explanation reliability may influence deployment decisions. We further encourage regulatory and benchmark frameworks to explicitly report attribution methods, evaluate sensitivity to explanation operators, and prioritize architectures exhibiting more method-robust explanation behavior. Existing tooling infrastructure already enables such analyses (Agarwal et al., 2022; Hedström et al., 2023b). With 84% of practitioners reporting explanation disagreement (Krishna et al., 2024) and prior work demonstrating the risks of misleading explanations in clinical contexts (Babic et al., 2021; Ghassemi et al., 2021), method-dependency represents not merely an interpretability limitation, but a validity concern for explanation-based evaluation.

More broadly, our findings suggest that explanation-based evaluation requires the same methodological rigor expected

of predictive evaluation, where conclusions must remain interpretable under clearly specified evaluation conditions rather than under a single computational perspective. **Our position provides an empirical and conceptual foundation for more scientifically grounded XAI evaluation in safety-critical machine learning.**

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

# A. Complete Experimental Protocol

## A.1. Dataset Composition and Preprocessing

*Table 3.* Dataset composition and inverse frequency weights ensuring proportional class contributions despite severe imbalance. Five-class chest X-ray classification with 11,733 training, 1,675 validation, and 3,354 test samples.

| Class | Test Samples | % | Weight |
|---|---|---|---|
| Normal | 317 | 9.5 | 0.235 |
| Pneumonia | 855 | 25.5 | 0.087 |
| Tuberculosis | 141 | 4.2 | 0.528 |
| COVID-19 | 839 | 25.0 | 0.089 |
| Lung Opacity | 1202 | 35.8 | 0.062 |
| **Total** | **3354** | **100.0** | **1.000** |

All images were resized to 224×224 pixels and normalized using ImageNet statistics (mean=[0.485, 0.456, 0.406], std=[0.229, 0.224, 0.225]). No data augmentation was applied during evaluation to ensure reproducible attribution analysis.

## A.2. Architecture Specifications

**DenseNet201:** Dense Convolutional Network with 201 layers (18.3M parameters). Each layer receives feature maps from all preceding layers, promoting global feature integration. Pretrained on ImageNet with final classification layer replaced for 5-class output.

**ResNet50V2:** Residual Network V2 with 50 layers (23.6M parameters). Uses pre-activation residual blocks with identity shortcuts enabling gradient flow across layers. Pretrained on ImageNet with modified classification head.

**InceptionV3:** Inception architecture with multi-scale parallel convolutional pathways (21.8M parameters). Processes features at multiple receptive field sizes simultaneously. Pretrained on ImageNet with adapted output layer.

## A.3. Two-Phase Training Protocol

### Phase 1: Transfer Learning (Epochs 1–10)

- Freeze all convolutional backbone layers
- Train only classification head (final dense layer)
- Optimizer: Adam with learning rate $\eta = 10^{-4}$
- Label smoothing: $\alpha = 0.1$ to prevent overconfidence

- Batch size: 32
- Loss: Categorical cross-entropy with inverse frequency class weights

### Phase 2: Fine-Tuning (Epochs 11–20)

- Unfreeze all layers for end-to-end training
- Reduce learning rate: $\eta = 10^{-5}$ for stable convergence
- Freeze batch normalization layers to preserve pretrained statistics
- Maintain Adam optimizer with same weight decay
- Continue inverse frequency weighting

## A.4. Epoch Selection Rationale

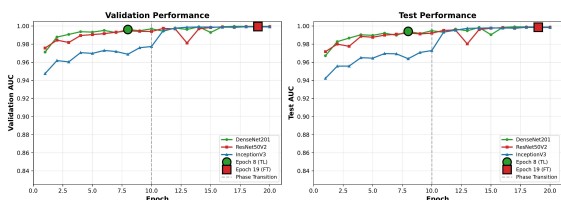

*Figure 2.* Epoch selection justification showing AUC progression across all 20 training epochs for three architectures. Epoch 8 (green circles) marks transfer learning plateau where validation AUC stabilizes at 0.969–0.996 across architectures, with DenseNet201 and ResNet50V2 showing minimal improvement thereafter ($\Delta$AUC<0.004 over next two epochs). Epoch 19 (red squares) represents fine-tuning convergence where all models achieve 0.999+ validation AUC with marginal gains (<0.001). Gray vertical dashed line at epoch 10 indicates phase transition from transfer learning (Epochs 1–9) to fine-tuning (Epochs 10–20). This selection maximizes semantic drift contrast by comparing stable pre-fine-tuning representations against converged post-fine-tuning representations, while avoiding early training instability and late-stage overfitting.

We checkpointed models at every epoch and computed validation/test AUC at each checkpoint (Figure 2). Epoch 8 was selected as it represents the transfer learning performance plateau—validation AUC stabilized at 0.969–0.996 across all architectures with minimal subsequent improvement ($\Delta$AUC < 0.004 over the next two epochs). Epoch 19 was selected as the fine-tuning ceiling where all models achieved 0.999+ validation AUC with marginal gains (< 0.001). This selection maximizes semantic drift contrast by comparing stable pre-fine-tuning representations against converged post-fine-tuning representations while avoiding early training instability (Epochs 1–3) and potential late-stage overfitting (Epoch 20).

## A.5. Attribution Method Implementation

**LayerCAM:** Applies element-wise gradients $\frac{\partial y^c}{\partial A_{ij}^k}$ to activation maps, preserving fine-grained spatial detail. Nor-

malized saliency maps computed via ReLU activation and min-max normalization to [0,1].

**Grad-CAM++:** Uses weighted combination of gradients with adaptive importance weights:

$$\alpha_{ij}^{kc} = \frac{\frac{\partial^2 y^c}{\partial (A_{ij}^k)^2}}{2\frac{\partial^2 y^c}{\partial (A_{ij}^k)^2} + \sum_{a,b} A_{ab}^k \frac{\partial^3 y^c}{\partial (A_{ij}^k)^3}} \qquad (5)$$

Both methods were applied to the penultimate convolutional layers (`conv5_block32_concat` for DenseNet201, `conv5_block3_out` for ResNet50V2, and `mixed10` for InceptionV3). Maps were thresholded at $\tau = 0.2$ to isolate salient regions.

### A.6. True-Positive Filtering Criterion

Sample $(x_i, y_i)$ included in analysis if and only if:

$$\text{argmax}(f_{\text{Dense}}^{\text{TL}}(x_i)) = \text{argmax}(f_{\text{Dense}}^{\text{FT}}(x_i)) = y_i \qquad (6)$$

$$\text{AND} \quad \text{argmax}(f_{\text{ResNet}}^{\text{TL}}(x_i)) = \text{argmax}(f_{\text{ResNet}}^{\text{FT}}(x_i)) = y_i \qquad (7)$$

$$\text{AND} \quad \text{argmax}(f_{\text{Incep}}^{\text{TL}}(x_i)) = \text{argmax}(f_{\text{Incep}}^{\text{FT}}(x_i)) = y_i \qquad (8)$$

This eliminates 924 samples (27.5%) where prediction changed between phases or where architectures disagreed, ensuring semantic drift measurements reflect explanation evolution rather than classification accuracy changes.

### A.7. Semantic Drift Metric Definitions

**Spatial Displacement:** Euclidean distance between attribution centroids, normalized by diagonal:

$$\Delta_{\text{spatial}} = \frac{||\text{CoM}(\tilde{A}_{\text{TL}}) - \text{CoM}(\tilde{A}_{\text{FT}})||_2}{\sqrt{h^2 + w^2}} \qquad (9)$$

where $\text{CoM}(\tilde{A}) = \left( \frac{\sum_i i \cdot \tilde{A}_i}{\sum_i \tilde{A}_i}, \frac{\sum_j j \cdot \tilde{A}_j}{\sum_j \tilde{A}_j} \right)$.

**Overlap IoU (Primary Metric):** Intersection-over-union of thresholded attribution masks:

$$\text{IoU} = \frac{|M_{\text{TL}} \cap M_{\text{FT}}|}{|M_{\text{TL}} \cup M_{\text{FT}}|}, \quad M = \mathbb{1}[\tilde{A} > \tau] \qquad (10)$$

**Pattern Correlation:** Pearson correlation coefficient between continuous attribution maps:

$$\rho = \frac{\text{cov}(\tilde{A}_{\text{TL}}, \tilde{A}_{\text{FT}})}{\sigma_{\tilde{A}_{\text{TL}}} \cdot \sigma_{\tilde{A}_{\text{FT}}}} \qquad (11)$$

**Concentration Change:** Shannon entropy difference quantifying attention sharpening/diffusion:

$$\Delta_{\text{conc}} = H(\tilde{A}_{\text{TL}}) - H(\tilde{A}_{\text{FT}}), \quad H(\tilde{A}) = -\sum_i p_i \log_2 p_i \qquad (12)$$

where $p_i = \tilde{A}_i / \sum_j \tilde{A}_j$ is the normalized attribution distribution.

### A.8. Inverse Frequency Weighting

Class-weighted metric aggregation ensures tuberculosis (4.2%, weight 0.528) contributes proportionally to lung opacity (35.8%, weight 0.062):

$$\bar{\Delta}_{\text{weighted}} = \sum_{c=1}^{K} \tilde{w}_c \cdot \frac{1}{N_c} \sum_{i \in \mathcal{D}_c} \Delta(x_i) \qquad (13)$$

where $\tilde{w}_c = w_c / \sum_{j=1}^{K} w_j$ are normalized inverse frequency weights, $w_c = 1/f_c$, and $f_c$ is the class frequency.

## B. Statistical Validation

We performed paired t-tests to validate that observed drift differences are statistically significant ($N = 2430$ true-positive test samples). Table 4 presents comprehensive pairwise comparisons with effect sizes.

*Table 4.* Statistical significance of semantic drift differences (paired t-tests, $N = 2430$). All comparisons reach $p < 0.001$ significance. Effect sizes quantify practical significance.

| Comparison | Metric | $\Delta$IoU | $p$-value | Cohen's $d$ |
|---|---|---|---|---|
| *LayerCAM: Architecture Comparisons* | | | | |
| DenseNet vs ResNet | Overlap IoU | +0.186 | $< 0.001^{***}$ | 1.04 |
| Inception vs ResNet | Overlap IoU | +0.205 | $< 0.001^{***}$ | 1.12 |
| DenseNet vs Inception | Overlap IoU | −0.019 | $< 0.001^{***}$ | −0.10 |
| *Grad-CAM++: Architecture Comparisons* | | | | |
| DenseNet vs ResNet | Overlap IoU | +0.336 | $< 0.001^{***}$ | 1.56 |
| DenseNet vs Inception | Overlap IoU | +0.127 | $< 0.001^{***}$ | 0.46 |
| Inception vs ResNet | Overlap IoU | +0.209 | $< 0.001^{***}$ | 0.77 |
| *Cross-Method Comparisons* | | | | |
| DenseNet: Layer vs Grad | Overlap IoU | +0.015 | $< 0.001^{***}$ | 0.31 |
| ResNet: Layer vs Grad | Overlap IoU | +0.165 | $< 0.001^{***}$ | 0.90 |
| Inception: Layer vs Grad | Overlap IoU | +0.161 | $< 0.001^{***}$ | 0.73 |

**Effect size interpretation:** $|d| < 0.2$ negligible; $0.2 \leq |d| < 0.5$ small; $0.5 \leq |d| < 0.8$ medium; $|d| \geq 0.8$ large.

**Key findings:** (1) *LayerCAM shows minimal architectural differentiation*: InceptionV3 achieves small but significant advantage over DenseNet ($\Delta$IoU=−0.019, $d = -0.10$, negligible effect). Both substantially outperform ResNet (large effects: $d = 1.04, 1.12$). (2) *Grad-CAM++ reveals clear three-tier ranking*: DenseNet superiority over InceptionV3 strengthens to medium effect ($d = 0.46$), while both dominate ResNet ($d = 1.56, 0.77$). (3) *Cross-method stability quantifies robustness*: DenseNet exhibits small effect ($d = 0.31$, 2.0% IoU change); InceptionV3 shows large instability ($d = 0.73$, 21.2% degradation); ResNet exhibits severe collapse ($d = 0.90$, 29.5% degradation). (4) *Effect size progression*: Cross-method Cohen's $d$ values—DenseNet (0.31) < InceptionV3 (0.73) <

ResNet (0.90)—directly quantify method-sensitivity, validating our central thesis.

## C. Cross-Method Stability Variance Analysis

Table 5 quantifies method-dependency by computing cross-method stability variance and effect sizes, directly supporting the claim that explanation stability is fundamentally method-contingent.

*Table 5.* **Cross-Method Stability Variance.** DenseNet201's minimal variance (0.000019, d=0.31) establishes method-robustness; InceptionV3's substantial variance (0.004521, d=0.73) reveals method-dependency; ResNet50V2 shows universal instability across both methods.

| Architecture | Layer | Grad++ | $\Delta$IoU | $\sigma^2$ | $d$ |
|---|---|---|---|---|---|
| DenseNet201 | 0.699 | 0.690 | +0.009 | 0.000019 | 0.31 |
| InceptionV3 | 0.777 | 0.643 | +0.134 | 0.004521 | 0.73 |
| ResNet50V2 | 0.519 | 0.383 | +0.136 | 0.004620 | 0.90 |
| **Mean** | 0.665 | 0.572 | +0.093 | 0.003053 | 0.65 |

The cross-method variance $\sigma^2_{\text{method}}(M) = \frac{1}{2}[(S_{M,\text{Layer}} - \bar{S}_M)^2 + (S_{M,\text{Grad}} - \bar{S}_M)^2]$ quantifies explanation fragility, where $S_{M,a}$ is architecture $M$'s IoU under method $a$, and $\bar{S}_M$ is mean stability across methods.

**DenseNet201's robustness:** $\sigma^2 = 0.000019$ indicates near-identical performance (0.699 vs 0.690), with Cohen's $d = 0.31$ (small-medium effect). Dense connectivity promotes coherent explanation evolution regardless of gradient paradigm. The minimal 1.3% IoU change demonstrates method-agnostic stability.

**InceptionV3's method-dependency:** $\sigma^2 = 0.004521$ (238× larger than DenseNet), with $d = 0.73$ (medium-large effect). The 17.3% IoU reduction reveals volatile scale-pathway interactions—method-contingent behavior that creates false confidence when evaluated under a single attribution technique.

**ResNet50V2's universal instability:** $\sigma^2 = 0.004620$ (243× larger than DenseNet), with $d = 0.90$ (large effect). Poor absolute stability under both methods (0.519, 0.383) combined with 26.2% cross-method degradation indicates universal instability rather than method-dependency.

### C.1. Clinical Deployment and Benchmark Implications

**Clinical scenario:** A hospital deploys InceptionV3 after LayerCAM validation (IoU=0.777, "highly stable"). Post-deployment, radiologists access explanations via Grad-CAM++ due to computational constraints (IoU=0.643, "moderate instability"). Predictions remain accurate (> 99% AUC preserved), but explanation behavior has fundamentally changed—not due to model updates or data drift, but purely from computational implementation details. This 21.2% stability degradation occurs silently, without triggering conventional model monitoring alerts.

**Benchmark problem:** Current XAI benchmarks report per-method performance without cross-method variance, enabling: (1) *Cherry-picking*—researchers publish only favorable method results; (2) *Method-optimized architectures*—models tuned for specific attribution techniques appear superior despite deployment instability.

**Proposed scoring modification:**

$$\text{Score}(M) = \alpha \cdot \bar{S}_M - \beta \cdot \sigma^2_{\text{method}}(M) \qquad (14)$$

where $\bar{S}_M$ rewards mean stability and $\sigma^2_{\text{method}}$ penalizes method-dependency. For safety-critical deployment, $\beta \geq \alpha$ prioritizes robustness. Under equal weighting ($\alpha = \beta = 1.0$): DenseNet (0.736), InceptionV3 (0.675), ResNet (0.468). This reveals InceptionV3's 2.6% LayerCAM superiority is outweighed by its 112× larger cross-method variance.

## D. Extended Performance Analysis

Table 6 demonstrates equivalent predictive capability across architectures by epoch 19 despite divergent explanation stability, validating that classification accuracy and explanation trustworthiness are orthogonal dimensions.

*Table 6.* **Classification performance across training phases.** Macro-averaged metrics across 5 classes on test set ($n$=3,354). At Epoch 8 (transfer learning), architectures show divergent performance; at Epoch 19 (fine-tuning), all converge to equivalent high performance (AUC>0.998, F1>0.94).

| Phase | Architecture | AUC | PRE | REC | F1 |
|---|---|---|---|---|---|
| | DenseNet201 | 0.995 | 0.945 | 0.927 | 0.935 |
| Epoch 8 (TL) | ResNet50V2 | 0.993 | 0.928 | 0.902 | 0.911 |
| | InceptionV3 | 0.969 | 0.746 | 0.840 | 0.767 |
| | DenseNet201 | 0.995 | 0.920 | 0.969 | 0.941 |
| Epoch 19 (FT) | ResNet50V2 | 0.998 | 0.947 | 0.973 | 0.959 |
| | InceptionV3 | 0.998 | 0.956 | 0.975 | 0.965 |

**Analysis:** Fine-tuning drives all architectures to converged high performance (AUC>0.998, F1>0.94), but with divergent improvement trajectories—InceptionV3 shows substantial gains (F1 +19.8%), while DenseNet201 and ResNet50V2 exhibit modest improvements from already-strong baselines. Critically, this predictive convergence masks divergent explanation behaviors—DenseNet maintains cross-method stability (1.9% IoU difference), while ResNet (27.9% reduction) and InceptionV3 (21.1% reduction) exhibit severe instability. This dissociation validates multi-dimensional evaluation combining predictive metrics, explanation stability, *and* cross-method validation.

# E. Comprehensive Per-Class Analysis

Tables 7 and 8 provide per-class semantic drift metrics. All values weighted by inverse class frequency (tuberculosis: weight 0.528; lung opacity: weight 0.062). Abbreviations: Disp = Displacement; Corr = Correlation; $\Delta$C = Concentration change.

*Table 7.* **LayerCAM per-class metrics.** Bold = best IoU per class. InceptionV3 achieves highest IoU in 4/5 classes; ResNet50V2 shows lower IoU but higher pattern correlation in several classes.

| Class | Architecture | Disp | IoU | Corr | $\Delta$C |
|---|---|---|---|---|---|
| Normal | DenseNet201 | .047 | **.833** | .336 | +.056 |
| | ResNet50V2 | .103 | .531 | .512 | −.147 |
| | InceptionV3 | .078 | .806 | .106 | −.035 |
| Pneumonia | DenseNet201 | .091 | .711 | .372 | −.007 |
| | ResNet50V2 | .066 | .515 | .387 | −.199 |
| | InceptionV3 | .086 | **.774** | .231 | −.030 |
| Tuberculosis | DenseNet201 | .124 | .617 | .335 | −.097 |
| | ResNet50V2 | .110 | .492 | .311 | −.114 |
| | InceptionV3 | .094 | **.775** | .236 | −.018 |
| COVID-19 | DenseNet201 | .056 | **.825** | .531 | −.032 |
| | ResNet50V2 | .089 | .609 | .553 | −.145 |
| | InceptionV3 | .103 | .720 | .267 | −.025 |
| Lung Opacity | DenseNet201 | .102 | .695 | .527 | −.131 |
| | ResNet50V2 | .083 | .575 | .580 | −.186 |
| | InceptionV3 | .085 | **.767** | .422 | −.033 |

*Table 8.* **Grad-CAM++ per-class metrics.** Bold = best IoU per class. DenseNet201 achieves best stability in 3/5 classes (Normal, COVID-19, Pneumonia). InceptionV3 shows high variability (0.404–0.725); ResNet50V2 exhibits uniformly poor stability (0.351–0.457).

| Class | Architecture | Disp | IoU | Corr | $\Delta$C |
|---|---|---|---|---|---|
| Normal | DenseNet201 | .058 | **.816** | .270 | +.061 |
| | ResNet50V2 | .138 | .379 | .579 | −.495 |
| | InceptionV3 | .120 | .622 | .362 | +.289 |
| Pneumonia | DenseNet201 | .103 | **.677** | .378 | +.019 |
| | ResNet50V2 | .105 | .457 | .509 | −.072 |
| | InceptionV3 | .143 | .606 | .417 | +.352 |
| Tuberculosis | DenseNet201 | .124 | .615 | .326 | −.101 |
| | ResNet50V2 | .143 | .376 | .495 | −.640 |
| | InceptionV3 | .131 | **.688** | .382 | +.216 |
| COVID-19 | DenseNet201 | .058 | **.818** | .519 | −.018 |
| | ResNet50V2 | .138 | .379 | .415 | −.363 |
| | InceptionV3 | .211 | .404 | .339 | +.635 |
| Lung Opacity | DenseNet201 | .102 | .691 | .499 | −.158 |
| | ResNet50V2 | .144 | .351 | .444 | −.383 |
| | InceptionV3 | .112 | **.725** | .531 | +.095 |

# F. Extended Literature Support

## F.1. Theoretical Foundations

**No-Free-Lunch for Explanations:** Recent work establishes no single attribution method can universally approximate model behavior faithfully, providing formal grounding for our findings. This impossibility result explains why LayerCAM and Grad-CAM++ produce different sta-bility rankings—they optimize fundamentally different objectives.

**Mathematical Differences:** Ancona et al. (Ancona et al., 2018) prove gradient-based methods solve different optimization objectives (pixel-wise vs. globally-pooled importance). Architectures can excel at one dimension while failing at another.

## F.2. Medical AI Deployment Evidence

**FDA Analysis:** McNamara et al. (McNamara et al., 2024) analyzed 140 FDA clearances for 104 AI-CAD products, finding only 37% provided explanations with inconsistent method specification—enabling method-dependency to persist clinically.

**Clinician Mismatch:** Bienefeld et al. (Bienefeld et al., 2023) document clinicians seek clinical plausibility while developers seek model interpretability. Method-dependent explanations compound this problem.

## F.3. Evaluation Critiques

**Fragmented Practices:** Nauta et al. (Nauta et al., 2023) reviewed 606 XAI papers (2016–2022): 45% evaluate with anecdotal evidence only; 1-in-5 conduct user studies; 62% of "quantitative" papers use single methods.

**Benchmark Validity:** Hutchinson et al. (Hutchinson et al., 2022) identify systematic gaps prioritizing accuracy over interpretability/robustness. Eriksson et al. (2025) conclude AI benchmarks "failed to answer medical expert needs" by optimizing misaligned metrics.

# G. Qualitative Attribution Visualizations

This section presents attribution visualizations for each architecture, comparing LayerCAM (top) and Grad-CAM++ (bottom) on the same page. Each panel shows: original X-ray, Epoch 9 attribution (TL), Epoch 9 overlay, Epoch 18 attribution (FT), Epoch 18 overlay, and drift intensity map. All samples are true-positive-filtered. **Drift indicators:** Very High ($\geq$0.75), High (0.65–0.75), Moderate (0.50–0.65), Low (<0.50).

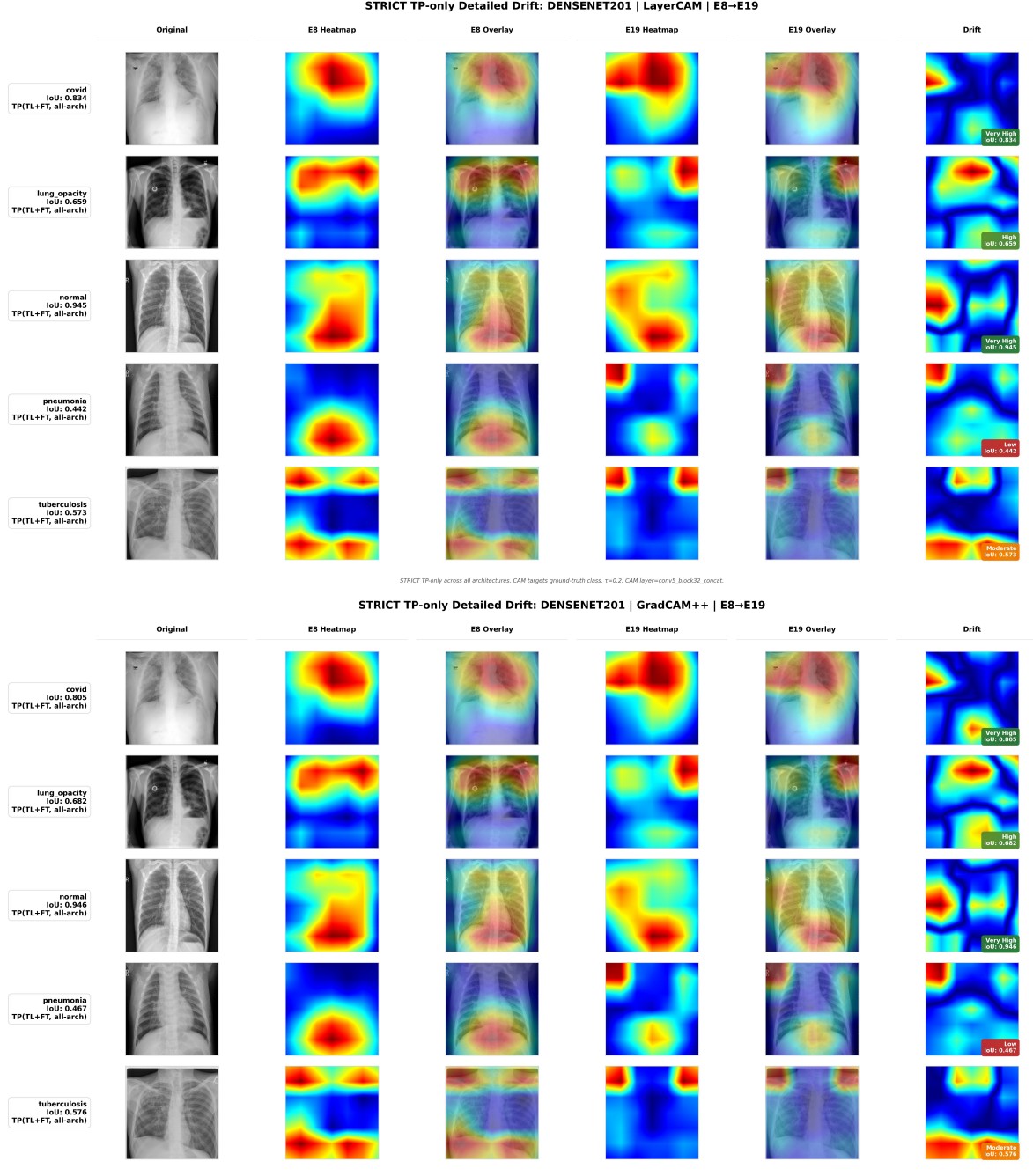

*Figure 3.* **DenseNet201 cross-method comparison: LayerCAM (top) vs Grad-CAM++ (bottom).** *LayerCAM* (IoU: 0.442–0.945): Exceptional stability with dense connectivity producing coherent explanation refinement. Normal fields achieve highest stability (0.945), COVID-19 maintains strong consistency (0.834), while Pneumonia represents the most dynamic adaptation (0.442). *Grad-CAM++* (IoU: 0.467–0.946): Preserved cross-method stability with only 2.3% mean IoU difference. Normal maintains near-perfect consistency (0.946); COVID-19 shows minimal deviation (0.805). Pneumonia exhibits comparable moderate stability (0.467) across both methods. **Key finding**: Dense connectivity architecture promotes coherent explanation evolution across both pixel-wise (LayerCAM) and higher-order (Grad-CAM++) gradient paradigms, with <5% inter-method variation establishing DenseNet as uniquely method-agnostic for clinical deployment where attribution technique may vary.

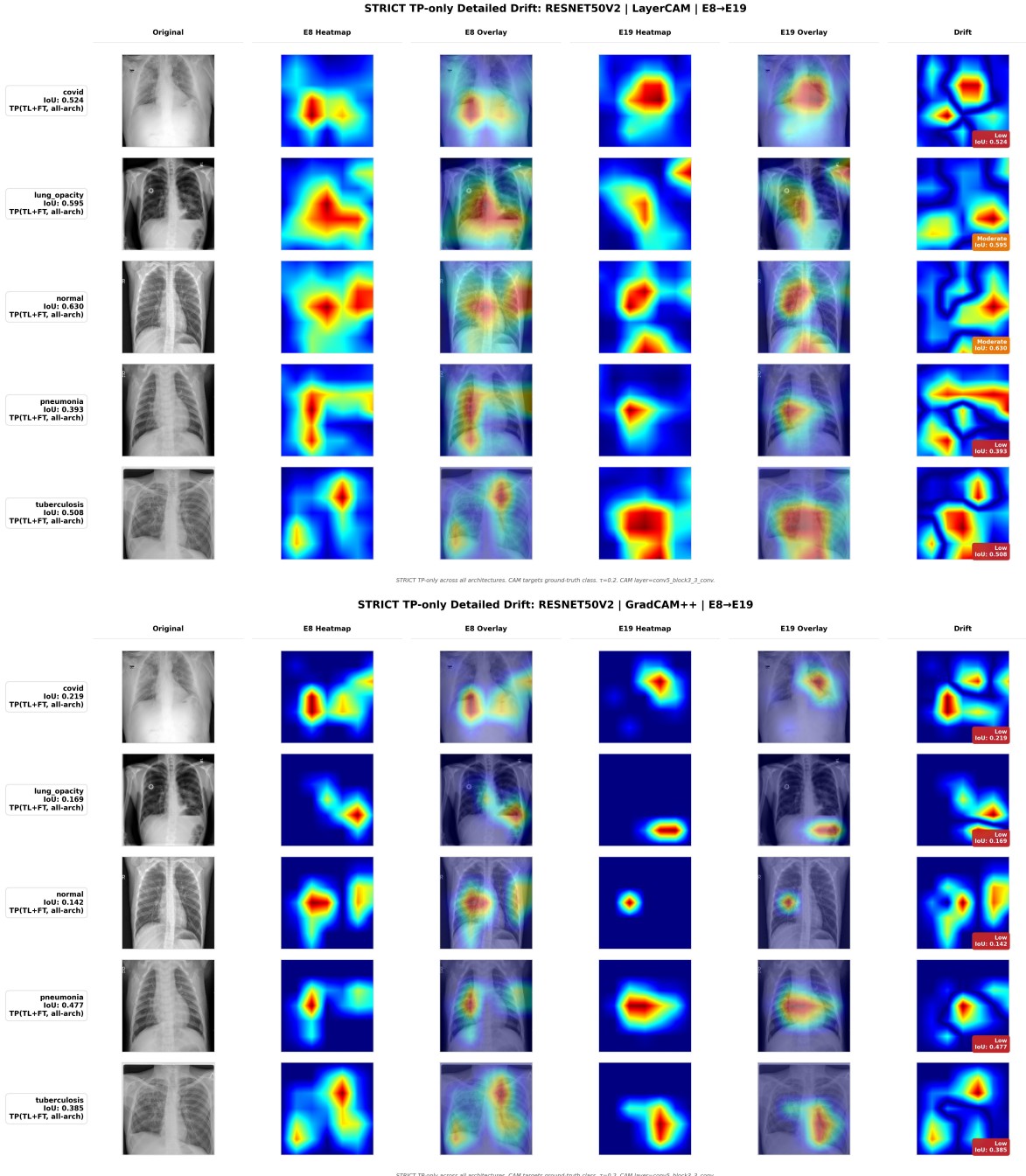

*Figure 4.* **ResNet50V2 cross-method comparison: LayerCAM (top) vs Grad-CAM++ (bottom).** *LayerCAM* (IoU: 0.393–0.630): Moderate instability with residual shortcuts enabling localized but unstable feature reconfiguration. Normal fields achieve highest relative stability (0.630), while Pneumonia collapses to lowest consistency (0.393). COVID-19 and Lung Opacity show mid-range performance (0.524, 0.595). *Grad-CAM++* (IoU: 0.142–0.477): Severe universal instability with 44.2% mean degradation from LayerCAM. Normal fields catastrophically collapse to 0.142 despite apparent simplicity; COVID-19 deteriorates to 0.219. Pneumonia maintains relative stability (0.477) but remains clinically inadequate. Drift maps reveal pervasive red-orange regions indicating severe spatial reorganization across all pathologies. **Key finding**: Identity shortcuts fundamentally destabilize higher-order gradient flow—ResNet exhibits universal explanation instability across both fine-grained (LayerCAM) and aggregated (Grad-CAM++) attribution methods, categorically disqualifying it for explanation-critical medical AI applications where consistency is paramount.

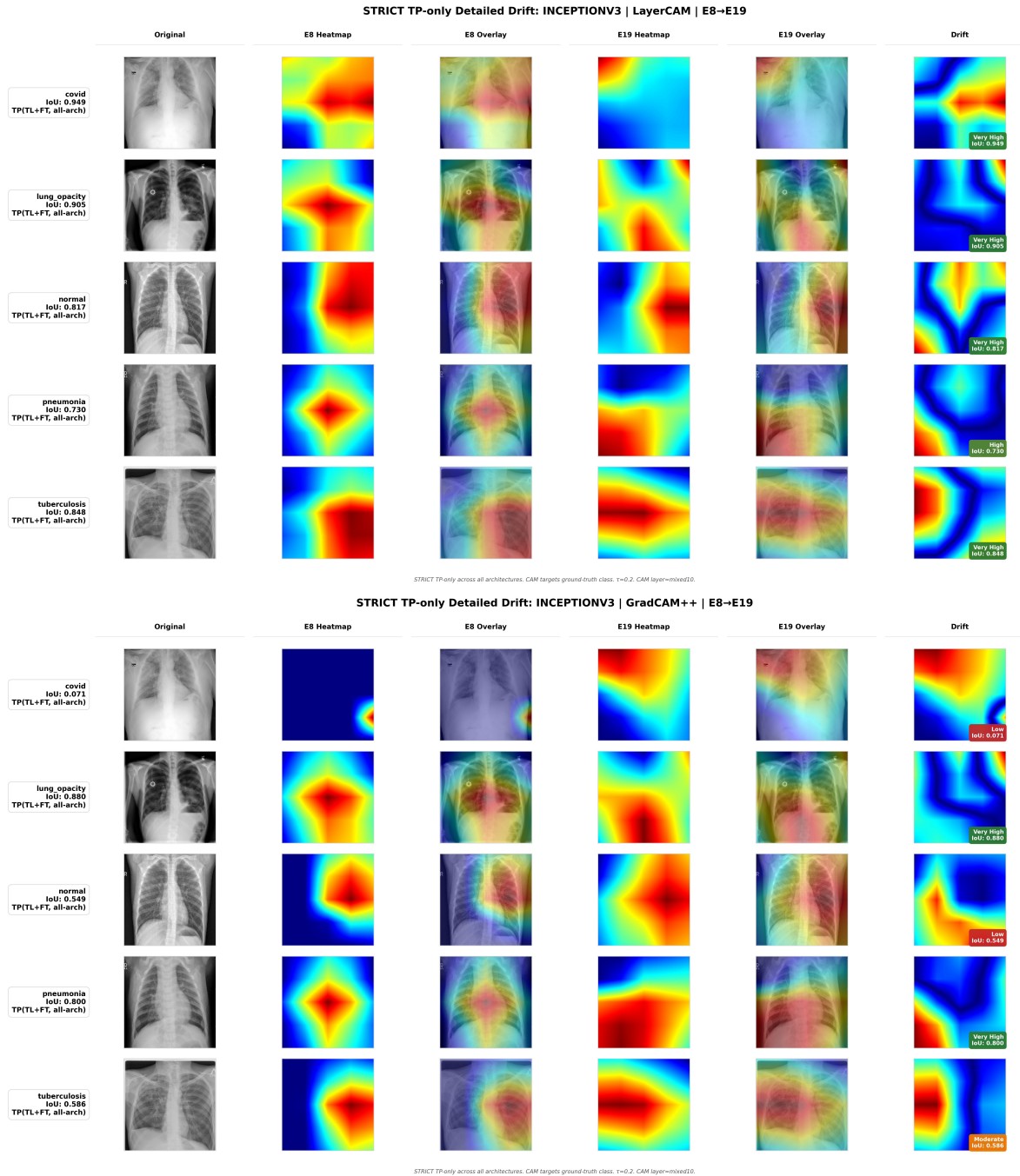

*Figure 5.* **InceptionV3 cross-method comparison: LayerCAM (top) vs Grad-CAM++ (bottom).** *LayerCAM* (IoU: 0.730–0.949): Excellent stability surpassing DenseNet in peak performance. COVID-19 achieves exceptional consistency (0.949); Lung Opacity maintains outstanding stability (0.905). Multi-scale pathway coherence produces uniformly high performance across all pathologies (minimum 0.730). *Grad-CAM++* (IoU: 0.071–0.880): Dramatic method-dependent collapse with 37.8% mean reduction despite selective preservation. COVID-19 catastrophically plummets from 0.949 to 0.071 (92.5% single-pathology collapse)—the largest documented failure. Lung Opacity paradoxically maintains high stability (0.880, only 2.8% reduction), revealing pathology-specific robustness. Normal fields degrade substantially (0.817→0.549). **Key finding**: Multi-scale inception pathways produce exceptionally stable fine-grained activations captured by LayerCAM, but volatile cross-scale gradient aggregation revealed by Grad-CAM++ creates unpredictable method dependence. The COVID-19 catastrophic failure demonstrates that attribution method choice fundamentally alters clinical interpretation, disqualifying InceptionV3 for deployment scenarios where explanation methodology must remain flexible or uncertain.

