# OpenReview forum: "Position: Explanation Stability Is a Property of the Model–Method Pair, Not the Model"
_ICML.cc/2026/Position_Paper_Track — ICML 2026 Position Paper Track regular_

### Official Review · Reviewer_q6Ri · 2026-03-04

**Significance:** 3
**Argument Clarity:** 3
**Rating:** 4
**Confidence:** 4

**Questions:**

See the weaknesses/questions above.

**Alternative Views Section:**

Yes

**Compliance With Llm Reviewing Policy A Conservative:**

Affirmed.

**Discussion Potential:**

3

**Final Justification:**

Rebuttals addressed my concerns. The paper emphasizes an important aspect of explainability but I feel this position may not be completely novel. I maintain my score.

**Paper Summary:**

This position paper argues that the claims of stability of an explanation method are only valid when the stability is tested across different methods. This paper shows that existing explanation methods are not stable when verified across different methods. Based on experiments on inception network, dense network and resnet architectures this paper shows that the different explanation methods such as LayerCAM and Grad-CAM++ show different stability patterns. The paper then proceeds to alternative views and argues for cross-method validation for explanation stability claims. The paper ends with suggestion to best practices in method selection and cross-method verification.

**Position:**

Yes

**Position In Title:**

Yes

**Related Work:**

3

**Strengths And Weaknesses:**

Strengths:

1. The paper is written very well and easy to understand.
2. Formalizing the best practices in explainable AI is an important problem to study.

Weaknesses/Questions:

1. In L64 right: what are "both training phases"? I think those terms are defined later in the paper.
2. In L133 left: it is said that "we argue practitioners need to identify which architectures exhibit method-robust stability". is this feasible considering many existing explanation methods?
3. There is no formal definition of "explanation stability" in the context of model-method pair. Since the context of model-method pair is a key problem setting introduced in this paper, it is good to have a formal definition in this context.
4. The position is taken based on a limited experimental setup (vision setting, three methods, two explanation methods) . While this shows a strong evidence, it only appears like an empirical observation. Also, how surprising is the position to the community? Are there any existing works that acknowledge this problem of explanation stability?

**Support:**

3

---

> ### Author Rebuttal · Authors · 2026-03-27
>
> We thank the reviewer for the constructive feedback and for highlighting areas where additional clarity can strengthen the manuscript.
>
> **Q1: Clarity (training phases)**
>
> We thank the reviewer for this observation. The “both training phases” refer to the transfer learning (frozen) and fine-tuning (unfrozen) stages described in Section 3.1. We will revise the manuscript to introduce and define these terms earlier for improved readability.
>
> **Q2: Feasibility of identifying method-robust architectures**
>
> We agree that evaluating all possible explanation methods is impractical. Our proposal is not exhaustive evaluation, but targeted validation across methodologically distinct explanation operators. In practice, testing across two representative and diverse explanation families (e.g., gradient-based and perturbation-based) can provide a practical and informative indication of whether a stability claim is robust or method-contingent.
>
> If stability does not hold across these distinct paradigms, the claim cannot be generalized to the model. Conversely, identifying architectures that remain consistent across such families provides a practical and actionable criterion for selecting more reliable models in safety-critical settings.
>
> **Q3: Definition of explanation stability**
>
> We agree that a more explicit definition will improve clarity. In the context of a model–method pair (M, A), we define explanation stability \( S(M, A) \) as the invariance of the attribution map \( A(M, x) \) under a set of model perturbations (in our study, those induced by fine-tuning).
>
> Concretely, this is quantified as the expected similarity between explanations generated before and after perturbation, using complementary metrics such as overlap IoU, spatial displacement, pattern correlation, and concentration change (Section 3.2). We will incorporate this formalization into the manuscript to more clearly ground the model–method interaction.
>
> **Q4: Novelty and prior work**
>
> We acknowledge that instability and disagreement across explanation methods have been previously reported. Our contribution is to formalize their implication as a constraint on the validity of explanation claims.
>
> Specifically, we argue that stability is not a model-intrinsic property, but a property of the model–method pair. This leads to a stronger conclusion: claims derived from single-method evaluation are not well-defined as general model properties. In this sense, we shift the perspective from treating instability as an empirical observation to treating it as a validity condition, motivating cross-method validation as a principled requirement rather than an optional best practice.
>
> We hope these clarifications address the remaining concerns and help strengthen the overall evaluation of the work.

---

> > ### Author Rebuttal · Reviewer_q6Ri · 2026-04-01
> >
> > Thank you for the response.
> >
> > If the instability and disagreement across explanation methods have been previously reported in literature, how are the findings/suggestions/position in this paper are novel? How does the existing work acknowledge this? Do they also provide alternative explanation validation techniques?

---

### Official Review · Reviewer_kRiL · 2026-03-05

**Significance:** 3
**Argument Clarity:** 3
**Rating:** 5
**Confidence:** 3

**Questions:**

- Q1: Could the authors elaborate on whether the method-dependent ranking reversals observed between gradient-based techniques persist when extending the framework to fundamentally different attribution families?

- Q2: Since current metrics only measure consistency rather than clinical validity, how should practitioners weigh method-robust stability against human-aligned correctness if a stable model remains consistently misaligned with radiologist reasoning?

**Alternative Views Section:**

Yes

**Compliance With Llm Reviewing Policy A Conservative:**

Affirmed.

**Discussion Potential:**

3

**Final Justification:**

After the rebuttal, the authors and I have reached an agreement on the practical importance of the proposed method. I consider it valuable and therefore support the acceptance of this manuscript.

**Paper Summary:**

This position paper argues that explanation stability is an emergent property of the model-method pair rather than an intrinsic model trait, rendering stability claims based on a single attribution method scientifically invalid. It contributes an empirical framework utilizing true-positive filtering to demonstrate that stability rankings among high-performing architectures reverse when switching attribution paradigms, establishing cross-method validation as a requirement for trustworthy deployment.

**Position:**

Yes

**Position In Title:**

Yes

**Related Work:**

3

**Strengths And Weaknesses:**

Strengths

- S1: The paper’s central position is forcefully substantiated by the high degree of consistency between its current experimental evidence regarding semantic drift and established literature on the pervasive disagreement problem in XAI.

- S2: Through the implementation of true-positive filtering, the study successfully isolates explanation evolution from changes in prediction quality, ensuring that measured instability reflects genuine shifts in internal representations rather than mere error correction.


Weaknesses

- W1: While the study effectively illustrates ranking reversals, the empirical evidence is primarily restricted to two gradient-based methods (LayerCAM and Grad-CAM++), leaving the generalizability of these findings across other families largely unexplored.

- W2: The current metrics quantify structural and spatial consistency but do not integrate human-expert annotations; consequently, it remains unclear if a method-robust architecture like DenseNet also aligns more closely with actual clinical correctness as judged by radiologists.

**Support:**

3

---

> ### Author Rebuttal · Authors · 2026-03-27
>
> We thank the reviewer for the insightful feedback and for highlighting the alignment between our findings and the broader literature on explanation disagreement.
>
> **Q1: Generalization beyond gradient-based methods**
>
> We agree that our empirical evaluation is currently limited to gradient-based methods. However, our central argument is not tied to this specific choice, but to the broader existence of non-equivalent explanation operators.
>
> LayerCAM and Grad-CAM++ both rely on gradients of the same feature maps, yet we observe ranking reversals between them. This suggests that the disagreement arises not from implementation details, but from how different methods formalize “importance.” More fundamentally, different explanation families encode distinct mathematical priors (e.g., local sensitivity, perturbation-based marginal contribution, or concept-level attribution). Prior work (e.g., Krishna et al., 2024) has shown that these families can produce contradictory explanations for the same prediction.
>
> Our contribution is to demonstrate, in a controlled setting where predictive performance is held constant, that model-level conclusions (e.g., stability rankings) are sensitive to these differences. We therefore view our findings as a lower bound on the disagreement problem: if reversals occur even within closely related gradient-based methods, they are likely to persist and potentially amplify across fundamentally different paradigms.
>
> Accordingly, our claim is not that all methods will disagree, but that single-method evaluation is insufficient to support model-level stability claims, as such claims remain conditional on the explanation operator.
>
> **Q2: Stability vs human-aligned correctness**
>
> We fully agree that stability alone does not imply correctness or alignment with expert reasoning. We instead position cross-method stability as a necessary but not sufficient condition analogous to reliability in measurement theory: consistency must precede meaningful assessment of correctness.
>
> In medical imaging, explainable AI is used to verify that predictions are grounded in clinically relevant features rather than spurious correlations. While this need is especially pronounced in this domain, similar concerns arise broadly wherever explanations are used to assess reliance on meaningful signal versus unintended shortcuts. In such settings, if explanations are unstable or method-dependent, it becomes unclear whether observed patterns reflect true model behavior or artifacts of the explanation operator.
>
> This issue is compounded by a mismatch in evaluation granularity: model predictions are typically image-level, whereas clinical annotations are often lesion-level, making direct alignment difficult. Moreover, expert annotations are costly and not scalable, so attribution methods are commonly used as practical proxies for model reasoning. In this context, instability is problematic regardless of correctness, as it indicates that the evidential basis of a prediction is not consistent even when predictions remain unchanged. Our use of true-positive filtering ensures that this reflects genuine shifts in explanation structure rather than prediction errors.
>
> From a practical perspective:
>
> - **Instability as a red flag:** If a model appears stable under one method but unstable under another, the explanation cannot be reliably interpreted as reflecting the model’s reasoning.
> - **Stability enables diagnosis:** If a model is stable but misaligned with expert reasoning, this represents a consistent and diagnosable failure; if unstable, it becomes a silent failure where explanations themselves are unreliable.
>
> To support operationalization, we further propose in Section 6.4 a scoring formulation that incorporates cross-method variability as a penalty term, encouraging the selection of models that exhibit method-robust explanation behavior. This allows practitioners to prioritize models whose explanations are consistent across interpretability “lenses,” ensuring that subsequent human-alignment evaluation is grounded in stable model behavior rather than method-specific artifacts.
>
> We will clarify these distinctions between consistency and correctness more explicitly in the revised manuscript. We hope these clarifications address the remaining concerns and help strengthen the overall evaluation of the work.

---

> > ### Author Rebuttal · Reviewer_kRiL · 2026-04-04
> >
> > I appreciate the authors’ rebuttal and acknowledge their clarification regarding the necessary but not sufficient conditions. Accordingly, I will raise my score.

---

### Official Review · Reviewer_MG3A · 2026-03-12

**Significance:** 2
**Argument Clarity:** 4
**Rating:** 3
**Confidence:** 4

**Questions:**

Q: Do the authors believe that the model–method interaction framing proposed here has implications that extend *beyond* adding yet another instance to the XAI evaluation crisis literature? Specifically, does this position offer a meaningful reorientation in how the community should think about the validity of XAI evaluation frameworks more generally — rather than arriving at the familiar conclusion that current evaluation practices are unreliable? If so, making this broader significance more explicit could substantially strengthen the case for why this position merits dedicated discussion at the venue.

**Alternative Views Section:**

Yes

**Compliance With Llm Reviewing Policy A Conservative:**

Affirmed.

**Discussion Potential:**

2

**Final Justification:**

I believe that this paper successfully demonstrates one of the instability issue of XAI benchmark, which is model-method dependency. However, this kind of specific limitation of benchmark limit the scope of the discussion invoke by position paper, which makes me to hessitate to recommend this as accept. Therefore, I would like to recommend borderline reject for this paper.

(I will change the recommendation if the author resolve my concern with their additional rebuttal)

**Paper Summary:**

**Overview**

This paper identifies a fundamental problem in current XAI evaluation practice. Many studies observe strong results from a single attribution method under a single experimental setting, then immediately generalize, concluding that the model itself is *explainable* or that its explanations are *stable*. The authors argue this inference is flawed: explanation stability is not a property of a model in isolation, but an emergent property of the model × explanation method interaction. Results obtained from a single method cannot be generalized, and cross-method validation is therefore essential.

---

**Background: Failure Modes in Current XAI Evaluation**

The authors situate their argument within a broader landscape of known evaluation failures:

- Sanity check failures; a classical finding showing that some explanation methods can produce outputs entirely independent of model parameters
- Explanation disagreement; different attribution methods yield conflicting explanations for the same prediction
- Reproducibility / evaluation instability; rankings and conclusions reverse depending on the dataset, model, or evaluation metric used

The authors emphasize that in medical imaging, where models are deployed in real clinical settings, these are not merely academic concerns: they are direct threats to reliability and safety.

---

**Experimental Setup**

The paper evaluates three CNN architectures on chest X-ray classification, using LayerCAM and Grad-CAM++ as attribution methods. Two key methodological choices are notable:

- True-Positive Filtering: Only samples that are predicted correctly across all training stages are retained, ensuring that observed changes in explanations cannot be attributed to changes in prediction accuracy.
- Stability Metrics: Explanation stability is measured via spatial displacement, overlap IoU, pattern correlation, and concentration change.

---

**Key Results**

The experimental results directly support the paper's central claim. For the same model, both the absolute stability values and the *relative rankings across models* change depending on which attribution method is used.

> A model that appears most stable under LayerCAM may not be the most stable under Grad-CAM++.

This means that conclusions such as *"this model is more explainable"* or *"more stable"* are method-dependent and cannot be treated as general properties of the model itself. This phenomenon is even more pronounced in per-class analysis than in aggregate averages.

---

**Core Conclusions**

| # | Conclusion |
|---|---|
| 1 | Strong performance under a specific explanation method does not imply general robustness |
| 2 | Accuracy alone cannot serve as a proxy for explanation quality or stability, even models with similar performance can differ substantially in explanation stability |
| 3 | Explanation stability should be treated as a property of the model–method pair, not of the model alone |

---

**Proposed Action Items**

- Cross-method validation: XAI evaluation should include at least two distinct attribution methods to verify consistency
- High-stakes deployment: In domains such as medical AI, the choice of explanation method must be included in evaluation, reporting, and regulatory frameworks
- Benchmarks and leaderboards: Should reflect not only single-method performance, but also cross-method consistency as a first-class evaluation criterion

**Position:**

Yes

**Position In Title:**

Yes

**Related Work:**

4

**Strengths And Weaknesses:**

**Strengths**
- Clarity of Motivation and Problem Framing

The paper offers a well-articulated reframing of a longstanding evaluation problem in XAI. It makes a clear and consistent case that explanation stability is not an intrinsic property of a model, but rather an emergent property of the interaction between the model and the attribution method — and that explainability claims grounded in a single XAI method are therefore fundamentally limited. The paper's position is stated clearly and developed consistently throughout.

- Thorough Contextualization of Related Work

The paper provides a broad and well-organized review of prior work on the XAI evaluation crisis, explanation disagreement, and sanity check failures. It also makes a compelling case for why explanation reliability matters in practice, particularly in real-deployment settings such as medical imaging. This background effectively grounds the paper's core motivation.

- Experimental Design Aligned with the Central Claim

While the experiments are relatively straightforward, they are well-suited to validating the paper's central thesis, i.e., explanation stability is method-dependent. The finding that model rankings by explanation stability shift across attribution methods provides direct and intuitive empirical support for the main argument.

---

**Weaknesses**
- Limited Scope: Broader Impact on the ICML Community

The discussion is heavily centered on attribution-based XAI evaluation. From the perspective of the broader ICML community beyond XAI, the reach and impact of this work appears somewhat limited, which is a particularly relevant consideration for the position track, where the ability to catalyze wide-ranging discussion across subfields is an important criterion.

- ★ Discussion Value: Does This Position Open *New* Conversations?

For a position paper, the strength of a submission rests not only on the solidity of the position itself, but on whether it has the potential to *stimulate genuinely new discussion*. On this dimension, I have reservations. The instability of XAI evaluation and the fragility of existing benchmarks are concerns that have been actively discussed in the community for more than 8 years. The core message — that single-method evaluation is unreliable — is a recognizable refrain in this space, and it is not clear that framing it through the lens of cross-method validation opens a meaningfully new conversation, as opposed to reinforcing an already well-established XAI evaluation instability issue.

---

**Conclusion**

The paper is clearly written and addresses a real problem, but its contribution sits within a well-worn area of XAI evaluation critique. Judged by the standards of the position track — where the value of the discussion a paper generates is as important as the position it advances — this submission is difficult to recommend. The problem is recognized, the solution is incremental, and the deeper questions worth debating are not sufficiently surfaced. The work may be better suited for a main track venue, where its methodological contributions can be evaluated on their own terms.

**Support:**

4

---

> ### Author Rebuttal · Authors · 2026-03-27
>
> We sincerely thank the reviewer for the careful and thoughtful evaluation. We greatly appreciate the recognition of the paper’s clarity, structured argumentation, and alignment between empirical findings and the central claim. We address the concerns regarding novelty, broader impact, and discussion potential.
>
> **On whether the model–method interaction framing provides a meaningful reorientation beyond existing XAI critique:**
>
> We fully agree that instability and disagreement across attribution methods have been well documented in prior work (e.g., Adebayo et al., 2018; Kindermans et al., 2019). However, our contribution is not to restate these observations, but to formalize their implication as a constraint on the validity of explanation claims.
>
> Specifically, we argue that explanation stability is not a model-intrinsic property, but a property of the model–method pair. This leads to a stronger conclusion than prior work. Claims such as "this model is stable" or "this model is explainable" are not well-defined unless conditioned on the attribution method used.
>
> To support this, our experiments provide a minimal counterexample to current evaluation practice. We show that model stability rankings reverse across attribution methods despite near-identical predictive performance (>99% AUC across all architectures). For instance, InceptionV3 appears most stable under LayerCAM (IoU = 0.777), while DenseNet201 becomes most stable under Grad-CAM++, with InceptionV3 experiencing a substantial drop. Because predictive performance is controlled, this demonstrates that explanation-based conclusions are method-dependent rather than model-intrinsic.
>
> Beyond this conceptual framing, we also provide a concrete step toward operationalization. In Section 6.4, we introduce a formulation that incorporates cross-method variability as a penalty term in model selection, encouraging the identification of models that exhibit method-robust explanation behavior. This moves the contribution beyond critique by offering a practical mechanism to account for explanation variability in evaluation pipelines.
>
> Importantly, the role of this experiment is not to claim universal instability, but to demonstrate that single-method evaluation is insufficient to establish model-level stability claims. This positions our contribution as a validity argument rather than an empirical extension of prior instability findings.
>
> **On whether this opens a new discussion or reinforces an existing narrative:**
>
> We respectfully suggest that the key shift is from
> "explanations may be unstable"
> to
> "when is a stability claim valid?"
>
> This reframing introduces an evaluation principle: explanation claims require cross-method invariance to be considered well-defined. While prior work identifies instability, it does not explicitly formalize how this affects the interpretability of evaluation outcomes. Our work attempts to bridge this gap by linking disagreement directly to the validity of conclusions drawn from XAI evaluation.
>
> **On broader ICML relevance and discussion potential:**
>
> We believe this reframing has broader implications beyond attribution methods:
>
> 1. **Evaluation validity:** It raises the question of whether widely reported explanation metrics are meaningful if they are conditional on a single operator.
> 2. **Benchmark design:** It suggests that model rankings based on explanation stability may not be invariant across methods.
> 3. **Regulatory alignment:** Emerging frameworks (e.g., FDA explainability evaluation) assess properties such as consistency and fidelity within a single method. Our findings highlight an additional axis—method-level variability that is currently unaddressed.
>
> In medical imaging classification tasks, where pixel-level ground truth explanations are often unavailable, attribution methods are frequently used as proxies. In this setting, method-dependent variability directly affects how explanation reliability is interpreted, even when predictive performance remains unchanged.
>
> Our position, therefore, is not that explanations must agree, but that disagreement must be reported when explanation-based claims are made in high-stakes settings. This shifts the discussion from observing instability to defining when explanation-based conclusions can be considered valid.
>
> We will revise the manuscript to make this broader significance and positioning more explicit. We hope this clarifies that the contribution is not incremental, but a shift toward validity-aware evaluation of explainability, which we believe merits discussion at the venue. We respectfully ask the reviewer to reconsider their assessment in light of these clarifications.

---

> > ### Author Rebuttal · Reviewer_MG3A · 2026-04-03
> >
> > Thank you for the thoughtful rebuttal, I appreciate the clarification.
> >
> > > “when is a stability claim valid?”
> >
> > I agree that this is a meaningful reframing. Reflecting on your response, I realize that my initial interpretation may have reduced the paper to “yet another instability observation,” whereas your intent is closer to identifying conditions under which a specific class of stability claims can be considered valid. I'll adjust my score to **borderline accept**.
> >
> > That said, my concern shifts accordingly:
> > - If the contribution is framed around model-method stability, then the scope appears to be limited to a particular type of stability within XAI evaluation. This may narrow the audience and reduce its relevance beyond a specific subset of attribution-based benchmarks.
> > - More importantly, even if this particular instability (i.e., method dependence) is addressed through cross-method validation or related mechanisms, it does not necessarily imply that XAI evaluation becomes globally stable or reliable. Many other sources of instability remain unaddressed, which also limit the impact of this paper.
> >
> > From this perspective, I am unsure whether focusing on a specific benchmark-level issue rises to the level of a position paper that can broadly stimulate new discussion across the community. Therefore, I will keep leaning toward reject.

---

### Official Review · Reviewer_mD2Q · 2026-03-16

**Significance:** 3
**Argument Clarity:** 3
**Rating:** 4
**Confidence:** 3

**Questions:**

1.The empirical section studies two gradient-based attribution methods on CNN architectures. How strongly do the authors believe their main conclusion generalizes to perturbation-based or attention-based explanation methods？
2.The paper argues for mandatory cross-method validation, but what is the minimal operational standard the authors would recommend in practice? For example, how many methods are sufficient, and how should they be selected to ensure meaningful diversity rather than superficial redundancy?

**Alternative Views Section:**

Yes

**Compliance With Llm Reviewing Policy A Conservative:**

Affirmed.

**Discussion Potential:**

3

**Final Justification:**

By considering the authors' response, I would like to maintain the original score.

**Paper Summary:**

This paper argues that explanation stability should not be treated as an intrinsic property of a model, but rather as a property of the model–method pair. Focusing on explainability evaluation in safety-critical medical imaging, the authors critique current XAI practice for drawing broad stability conclusions from single attribution methods, despite known disagreement across methods. They support this position with a controlled chest X-ray study comparing three CNN architectures under LayerCAM and Grad-CAM++, showing that stability rankings can reverse depending on the explainer even when predictive performance is nearly identical. Based on these findings, the paper calls for cross-method validation in XAI research and more explicit benchmark and regulatory standards for explanation robustness.

**Position:**

Yes

**Position In Title:**

Yes

**Related Work:**

3

**Strengths And Weaknesses:**

Strength：
1.The paper states a clear, specific, and nontrivial position. The central claim is easy to identify and is framed in a way that is testable and relevant: explanation stability should not be treated as a model-intrinsic property without cross-method validation.
2.The position is supported by both literature synthesis and empirical evidence. A major strength is that the paper does not rely only on conceptual argument. It backs its claim with a controlled experiment showing that stability rankings can reverse across attribution methods even when predictive performance is nearly identical. This directly supports the main thesis.

Weakness：
1.The paper makes a broad claim about explanation stability in XAI, but the core empirical evidence is based on two gradient-based attribution methods and three CNN architectures on one chest X-ray task family. This is enough to demonstrate the possibility of the problem, but it is still limited as evidence for a strong general policy recommendation such as mandatory cross-method validation in all relevant XAI work.
2.The paper convincingly shows that single-method stability claims can be misleading in its setting. However, moving from that observation to a universal requirement for cross-method validation would be more persuasive if the paper more clearly defined when such validation is essential, how many methods are enough, and what counts as sufficiently diverse attribution paradigms.

**Support:**

3

---

> ### Author Rebuttal · Authors · 2026-03-27
>
> We thank the reviewer for the constructive and well-balanced feedback, and for recognizing the clarity, specificity, and empirical support of the central claim.
>
> ---
>
> ### Q1: Generalization beyond gradient-based attribution methods
>
> We agree that the empirical study focuses on two gradient-based methods. However, our core argument does not depend on this specific choice, but on the broader existence of non-equivalent explanation operators. Different explanation methods encode fundamentally different mathematical objectives (e.g., local pixel sensitivity, perturbation-based importance, or concept-level attribution) and therefore introduce distinct inductive biases.
>
> Prior work has demonstrated that such methods can yield divergent explanations for the same prediction. More formally, theoretical *no-free-lunch* perspectives for explanations (e.g., Han et al., 2022) suggest that no single method can universally capture all aspects of model behavior. Our contribution is to show, in a controlled setting, that even when predictive performance is held constant, model-level conclusions such as stability rankings can change depending on the explanation operator used.
>
> Importantly, we observe this effect even within a single family (LayerCAM vs.\ Grad-CAM++). This suggests that the phenomenon is not tied to a specific class of methods but arises from the broader structure of explanation techniques. Consequently, if disagreement already emerges within closely related methods, it is likely to persist or even intensify across fundamentally different paradigms (e.g., gradient-based vs.\ perturbation-based or concept-based approaches).
>
> Our claim is therefore not that all methods will disagree, but that single-method evaluation is insufficient to support model-level stability claims, as such claims remain conditional on the explanation operator.
>
> ---
>
> ### Q2: Operational standard for cross-method validation
>
> We appreciate this important point and agree that clearer guidance strengthens the contribution. Our intent is not to prescribe a fixed number of methods or impose a rigid evaluation protocol, but to highlight that explanation stability is inherently dependent on the choice of explanation operator.
>
> In this context, cross-method validation should be viewed as a sensitivity analysis rather than a strict requirement. The key question is whether a model-level claim (e.g., *``Model A is more stable than Model B''*) remains invariant under different, methodologically distinct explanation paradigms, or whether it is contingent on a specific formulation.
>
> From a practical standpoint, evaluating across a small number of methodologically diverse explanation families (e.g., gradient-based vs.\ perturbation-based or concept-based approaches) can serve as an informative probe of this dependency. However, the primary objective is not the number of methods used, but to assess whether observed stability reflects a property of the model or an artifact of the explanation operator.
>
> We hope this motivates a broader interpretation of evaluation. Our position is not that explanations must agree, but that disagreement should be reported when explanation-based claims are made, particularly in high-stakes domains such as healthcare AI where such claims may influence trust and downstream decisions.
>
> We will revise the manuscript to make this distinction clearer, emphasizing that cross-method validation is not a prescriptive rule, but a necessary consideration for interpreting explanation-based claims. We hope this clarification addresses the reviewer’s concern and clarifies the positioning of the work.

---

> > ### Author Rebuttal · Reviewer_mD2Q · 2026-04-03
> >
> > Thanks for the authors' detailed response; my concerns have been adequately addressed.

---

### Decision · Program_Chairs · 2026-04-30

**Decision:**

Accept (regular)

**Comment:**

The authors of this paper argue that explanation stability depends on both the model and the explainability metric, not just the model.  Applications to critical areas like the medical domain are given.  Authors call for cross-method validation in XAI research to more accurately rank models.

Reviewers questioned if the evidence was enough to support the claim, and when it might be warranted to do cross-method xai validation. Reviewers also questioned the novelty of the claims. Authors argue that the paper reframes conversations around instability to be more concrete and actionable. The clarity provided about model stability vs model-method stability is important.

There was some question about the novelty of the claims — instability and disagreement across explanation methods have been previously reported.  However the authors argue that formalizing this relationship is important and making clear the source of the instability is also important.